# Design and Proof of Concept of a Web-Based Questionnaire to Identify Patients at Risk for HIV and HCV Infection

**DOI:** 10.3390/biomedicines12081846

**Published:** 2024-08-14

**Authors:** Alejandro G. García-Ruiz de Morales, María Jesús Vivancos, Jorge Lázaro, Beatriz Romero Hernández, Beatriz Mateos, Pilar Pérez-Elías, Margarita Herrero Delgado, Laura Villanova Cuadra, Santiago Moreno, Javier Martínez-Sanz, María Jesús Pérez-Elías

**Affiliations:** 1Department of Infectious Diseases, Hospital Universitario Ramón y Cajal, IRYCIS, Carretera de Colmenar Viejo, Km 9.100, 28034 Madrid, Spain; 2CIBER de Enfermedades Infecciosas (CIBERINFEC), Instituto de Salud Carlos III, 28029 Madrid, Spain; 3Department of Medicine, University of Alcalá, 28801 Alcalá de Henares, Spain; 4CIBER of Epidemiology and Public Health (CIBERESP), 28029 Madrid, Spain; 5Department of Microbiology, Hospital Universitario Ramón y Cajal, IRYCIS, 28034 Madrid, Spain; 6Department of Gastroenterology, Hospital Universitario Ramón y Cajal, IRYCIS, 28034 Madrid, Spain; 7Centro de Salud García Noblejas, 28037 Madrid, Spain; 8Centro de Salud Mar Báltico, 28033 Madrid, Spain; 9Centro de Salud Cirajas, 28017 Madrid, Spain

**Keywords:** late diagnosis, HIV risk questionnaire, HCV, primary care, pre-exposure prophylaxis

## Abstract

Despite remarkable improvement in the human immunodeficiency virus (HIV) and hepatitis C virus (HCV) care continuum, the rate of late diagnosis of HIV and HCV in high-income countries remains unacceptably high. Testing relies mainly on primary care physicians’ identification of risk factors. We aimed to adapt an analogic to an online questionnaire to help HIV and HCV screening and perform a pilot study to assess its accuracy and acceptability. We used the Delphi method to adapt a previously validated analogical questionnaire to a user-friendly online tool. It aimed to identify participants who should be screened for HIV or HCV and those who should be referred for pre-exposure prophylaxis (PrEP). We then designed a proof-of-concept pilot study from July to October 2022 to test its feasibility and suitability for use on a larger scale and to assess its accuracy in identifying patients at risk for HIV or HCV or with indication for PrEP. The final questionnaire consisted of 37 questions. A total of 142 participants provided informed consent, and 102 completed the questionnaire: 41 random patients recruited at the primary care level, 10 participants recently diagnosed with HIV, 20 participants with HIV on follow-up, 21 participants from the PrEP program, and 10 patients diagnosed with HCV. The tool adequately indicated the need for testing in more than 98% of participants with confirmed HIV/HCV infections or in the PrEP program. Furthermore, it adequately assessed PrEP referral in 94% of participants already on PrEP or with known HIV infection. Participants were highly satisfied with the tool, and 98% of them recommended its use. A self-administered web-based tool to identify patients who should be tested for HIV or HCV or referred to PrEP could simplify patient selection and help reduce late diagnosis.

## 1. Introduction

Undiagnosed human immunodeficiency virus (HIV) infection remains a significant public health challenge in the world that hinders achieving the goals outlined by the United Nations AIDS Program (UNAIDS) for HIV elimination. These goals include a substantial reduction in new HIV infections, early diagnosis of those living with HIV, and comprehensive care for all infected individuals [1].

Recent studies identify late diagnosis as one of the main drivers of new HIV infections, as well as the main challenge to successfully control the HIV pandemic, as some European countries are on the verge of reaching the UNAIDS 95-95-95 goals before the expected 2030 deadline [1,2]. The rate of late diagnosis of HIV in Europe continues to rise, and according to the European Centre for Disease Prevention and Control (ECDC), it is close to 50% [3], while the number of diagnosed HCV infections in the world between 2016 and 2022 was only 36% of all patients living with HCV [4].

While HIV and hepatitis C virus (HCV) testing guidelines vary, ranging from universal opt-out screening to more targeted approaches based on individual risk factors, the core principle remains the same: identifying individuals at risk of HIV and HCV exposure or with relevant indicator conditions to facilitate timely diagnosis and intervention and reduce late presentation [5,6,7]. These targeted strategies emphasize recommending HIV and HCV testing to all individuals seeking healthcare who present with specific indicator conditions. In spite of this, the assessment of individual HIV and HCV risks presents numerous challenges in real-world clinical settings. Time constraints, complex consent processes, limited training of healthcare professionals, the reluctance of patients, and competing healthcare priorities are significant barriers to optimal testing and risk assessment [8,9].

Specifically, systematic evaluation of HIV and HCV risk of exposure and indicator conditions in routine clinical practice is often absent, leading to a high incidence of missed opportunities and multiple inequities in HIV and HCV testing [9,10,11,12,13,14]. This is particularly challenging in primary care settings with high patient volumes and limited resources, where conducting a comprehensive assessment of every patient seeking care, regardless of their presenting concerns, can be impractical and overwhelming.

Focusing on our prior work in this area [14,15], we sought to create a more practical and efficient approach, specifically a web-based tool for HIV and HCV screening in primary care in the Spanish language. We aimed to adapt a questionnaire and to perform a pilot study to test its performance and participant satisfaction. The questionnaire aims to—in an upcoming study—enhance diagnostic accuracy, address potential inequities in HIV and HCV detection, and identify suitable candidates for HIV pre-exposure prophylaxis (PrEP). This innovative tool can significantly improve HIV testing and risk assessment in primary care by simplifying the risk assessment process and integrating it into the existing workflows. Ultimately, this advancement could contribute to achieving the UNAIDS goals and improving public health outcomes.

## 2. Materials and Methods

### 2.1. Development of the Questionnaire

#### 2.1.1. Initial Selection of Questions

We used a modified Delphi expert consensus method to develop the questionnaire [16,17]. Initially, the researchers pre-selected all the questions from a previously developed analogical risk questionnaire successfully used to screen for HIV and HCV in primary care [14,18] and modified it to include the most up-to-date HIV and HCV diagnostic and PrEP guidelines from the Spanish Ministry of Health, the Centers for Disease Control and Prevention (CDC), and the ECDC [19,20,21].

The researchers identified 50 questions to achieve the questionnaire’s objectives and categorized them into eight groups: introductory and affiliation (5), sexual practices (14), drug use (6), PrEP indication (13), causal acquisition of HIV/HCV (2), indicator conditions for HIV infection (14 answers in one question), missed diagnoses (4), and tool evaluation (5) (Appendix A).

#### 2.1.2. Modified Delphi Method

Based on predefined selection criteria (Appendix A, p. 2), we recruited 19 skilled physicians to perform a modified Delphi method. We included eight primary care doctors, two gastroenterologists, and nine specialists in HIV and sexually transmitted infections (STIs); they evaluated the importance of each question. They provided anonymous opinions in an online questionnaire using the following choices: questionable, unnecessary, essential, and necessary, with suggested wording modifications. The experts also had a free-text field to suggest new questions.

#### 2.1.3. Evaluation of Responses

The initial responses from consulted specialists were evaluated and assessed based on the RAND/UCLA criteria [22], which classified each question into one of three options: (a) if five or fewer experts suggested an alternative to “Essential”, the response was interpreted as indicating agreement to maintain the question; (b) we defined the question as “disagreement” if six or more experts considered it unnecessary; and (c) in all other cases, the question was classified as neutral.

### 2.2. Determining Eligibility for Testing and Prep

According to the current HIV/HCV testing and PrEP guidelines [19,20,21], we modified the previously validated rule-based algorithm to identify patients eligible for HIV/HCV testing or assess PrEP initiation. To simplify the questionnaire outputs, when the algorithm flagged participants for testing for either HIV or HCV, they were tested for both. Appendix A (p. 14) presents the complete algorithm extracted from the Research Electronic Data Capture (REDCap). The algorithm classified patients into three categories. Participants and their primary care providers received online and email notifications according to their category.

Eligibility for HIV/HCV testing: Patients who were identified as needing HIV and HCV testing were asked to discuss their results with a primary care provider for appropriate testing and were offered a rapid diagnostic test or conventional serology (per physician’s decision) at their primary care facility.Eligibility for PrEP evaluation: Patients were divided into no PrEP needed, straight PrEP derivation, or individual evaluation for PrEP by a physician. Regardless, the patient received a notification with information on PrEP eligibility and was encouraged to discuss potential initiation with their primary care provider. If eligible participants agreed to start PrEP, they were referred to the PrEP outpatient clinic without further primary care involvement.No further action: These patients were deemed unlikely to benefit from routine HIV/HCV testing or PrEP based on the questionnaire. They received confirmation that no further action was required at that time.

### 2.3. Implementation of the Questionnaire: Pilot Study

We uploaded the questionnaire to a REDCap tool hosted at Ramón y Cajal Hospital servers, inside the Community of Madrid’s secure servers [23]. We made it available to the public via a QR code and a URL (https://redcap.link/proyectoatenea). We asked the main investigator from the project to randomly recruit participants for the pilot study in each primary care center.

We conducted a small-scale implementation study to determine whether the tool was suitable for broader use and its precision in identifying patients at risk for HIV, HCV, or PrEP indication. We, therefore, selected a sample from primary care and some positive controls who were already HIV-positive or HCV-positive or included in the PrEP program. Each participant signed an electronic informed consent before accessing to the questionnaire.

Given the main aim of this part of the study, the investigators decided to include at least 100 patients in the pilot study: 40 patients at primary care level (5 from each of the eight assigned primary care centres at Ramón y Cajal Hospital) who would serve as the target population, where the feasibility and time to complete the questionnaire would be most important; 60 patients who would serve as positive controls to preliminarily assess the accuracy of the questionnaire; 20 HIV patients diagnosed between six months and ten years ago (positive controls for HIV testing); 10 patients newly diagnosed with HIV (positive controls for HIV testing and PrEP); 10 patients in the PrEP program (positive controls for HIV testing and PrEP); and 20 patients with active HCV infection (positive controls for HCV testing). Patients from primary care were selected as the first patient between 18 and 65 years old to attend primary care in five given days. Regarding the positive controls, their physicians randomly selected the patients from their outpatient clinic to complete the questionnaire according to their clinical criteria.

Participants in the “eligible for HIV/HCV testing” group underwent an HIV and HCV rapid diagnostic test or conventional serology, as determined by their primary care physician. Those with HIV or HCV infection and those already on the PrEP program were excluded from testing. The anti-HIV 1/2 test WB/S/P (TürkLab Laboratories) served as the rapid HIV test, whereas the anti-HCV WB/S/P test (TürkLab Laboratories) was used for HCV. The reference virology lab performed the serological tests using an Anti-HCV II assay and HIV Ag/Ab Combo Reagent Kit on Abbott Alinity’s platform.

We calculated accuracy, sensitivity, and specificity analyses of the questionnaire for both HIV and HCV testing and for PrEP derivation. We used data from the latest HIV and HCV prevalence in Spain (0.3% and 0.22%, respectively) [24,25] to make assumptions on the data we could not extract from the results for the calculations (Appendix A, p. 16).

#### Statistical Analysis

REDCap was used for data collection, while Stata/MP 18.0 (StataCorp LP, College Station, TX, USA) was used to conduct a descriptive analysis of the baseline characteristics using frequency distributions.

## 3. Results

### 3.1. Results of the Delphi Method

Of the 50 questions, 46 received “agreement”, and four received “neutral” ratings (Appendix A, p. 3). We received ten suggestions for new questions, three of which were coincidental, and multiple minor corrections to the wording of the initial ones.

Of the four questions with neutral ratings, two corresponded to the “demographic category”, one to the “sexual practices assessment category”, and the last to the “assessment of the indication for PrEP”. The research team eliminated the one for “assessing the indication for PrEP” and one of the demographic questions (Appendix A, p. 9). The first one (use of intramuscular penicillin G) was redundant with another question, while the latter (ethnicity) added irrelevant information for the objectives. In addition, we added the suggested coincidental questions and made minor changes to the wording of the initial questions, as suggested. Given the high agreement in the first round and the little changes made to the initial version, the research team evaluated the modified questionnaire and approved it as the final version, which was sent back to the experts for their approval (Appendix A, p. 9).

### 3.2. Results of the Pilot Study

A total of 142 users were enrolled in the pilot study and signed an informed consent form; 102 (71.8%) completed the questionnaire. The participants’ mean age was 39 years; 80 (78.4%) were men, 65 (63.7%) were born in Spain (the remaining 35% were fluent Spanish speakers), 92 (90.1%) had a high educational level, and 81 (82%) were active workers. Table 1 shows the distribution of participants and their baseline characteristics per group. Appendix A display the sexual habits, drug use, previous STIs, and HIV indicator conditions.

#### 3.2.1. Indication for Testing and Prep

The tool efficiently identified almost all patients in the HIV, PrEP, and HCV groups for HIV/HCV testing. It only missed one individual in the HIV follow-up group. This male participant reported no unprotected sex, a single male sexual partner, and no drug use, STIs, or HIV indicator conditions.

Of the 21 participants already receiving PrEP, 19 (90.5%) qualified for direct PrEP derivation through the tool, while the remaining two qualified for individual assessment of PrEP derivation. Among the 30 participants who lived with HIV, the tool flagged 27 (90%) as candidates for PrEP initiation if they had not been living with HIV. Interestingly, all three missed individuals belonged to participants diagnosed more than six months before (Table 2).

In the primary care cohort, the tool identified 26 participants (63.4%) for HIV/HCV testing (one HIV infection detected) and 9 participants (21.9%) as eligible for PrEP: 6 (14.6%) for direct derivation and 3 (7.3%) for individual evaluation.

The sensitivity, specificity, and accuracy of the questionnaire for identifying patients for HIV and HCV testing were 98.4%, 95.0%, and 97.1%, respectively, whereas for PrEP derivation, the results were 94.1%, 96.1% and 95.1%, respectively.

#### 3.2.2. Feasibility and Patient Reports

Most participants (100, 98%) found the questionnaire easy to answer and the tool easy to use, and 100 (98%) reported that they would recommend it to others. Only eight (7.8%) patients required external assistance to complete it. Most participants (96, 94.1%) used a personal smartphone to answer these questions. The median time to complete the questionnaire was 6 min 37 s (IQR: 4 min 43 s to 9 min 34 s) (Table 3).

## 4. Discussion

In this study, we demonstrate that an adapted up-to-date online questionnaire to improve HIV and HCV testing at the primary care level successfully identified the need for HIV/HCV testing in over 98% of the participants with confirmed infections or in a PrEP program, correctly advised referrals to PrEP for over 94% of patients already in the program or with a known HIV infection, and helped identify a high proportion of primary care participants as candidates for HIV/HCV screening. We aimed to improve diagnostic strategies to reduce the high rates of late diagnosis by adapting a previously validated analogical tool [15,18,26] to an online questionnaire for HIV/HCV testing and PrEP derivation in primary care settings.

Strategies such as universal opt-out HIV testing [27] and targeted screening through formative approaches in primary care [14,15,26] or hospital settings [9] improve HIV testing and diagnosis rates. However, assessing individual HIV or HCV risks presents significant challenges in real-world clinical settings, particularly for primary care providers, who still face barriers to offering testing to all at-risk patients [8]. Additionally, some medical personnel lack knowledge about PrEP, how to use it, and its benefits. This gap in testing and PrEP derivation is probably due to factors such as time constraints [28,29], staff shortages [30], and HIV stigma [31] and could be addressed using new tools to identify patients who should be tested.

Technologies have been successfully used to identify people at risk in various settings [32,33,34]. However, this process still relies on primary care physicians to accurately assess risk behaviours and diagnose indicator conditions. In high-income countries, most new HIV diagnoses occur in young adults [3] who are generally familiar with smartphones and online questionnaires. We propose that using an online tool can simplify the identification of at-risk patients and reduce the need for primary care physicians’ engagement for testing. This could address time constraints, patients’ reluctance to discuss specific conditions or sexual practices with their doctors, or the challenges physicians from older generations may face when talking about sexual health issues. Additionally, the study’s impact could be sustained in the long term by training healthcare professionals to use the tool effectively and by educating community stakeholders about the importance of HIV and HCV prevention and testing.

The pilot study showed that screening with the tool was feasible, easy to use, and had high participant satisfaction. With a small sample size, the questionnaire effectively identified patients at risk for HIV and HCV infections. Our future work will aim to distribute 10,000+ questionnaires at primary care facilities and assess whether the availability of such a tool is helpful for increasing HIV and HCV testing rates. If this is the case, the widespread use of such a tool could help reduce the percentage of hidden infections and late diagnoses.

The study has certain limitations. The tool’s effectiveness depends on the sincerity and involvement of patients and on some involvement of primary care providers, whereas some individuals may require external help to complete the questionnaire. However, this was not prevalent in the pilot study. Further, a certain level of education is needed to auto-complete an online questionnaire, which can make the implementation of the tool difficult among the most vulnerable. The main objective of the current study was not to calculate the accuracy, sensitivity, and specificity of the questionnaire, so these numbers should be received with caution. Nonetheless, a similar questionnaire used earlier by our group demonstrated similar sensitivity, specificity, and excellent predictive values [14,18]. We aim to confirm these results once we complete the next study phase. The pilot study was conducted in a limited number of participants who were mainly individuals with a high level of education. If this trend continues in the upcoming study, it may perpetuate the disparity in testing among the most vulnerable, and therefore, the tool might not be adequate for this population. The participants were selected randomly according to their physician’s criteria, except for those recruited in primary care. As such, some grade of selection bias could be present, although we do not think this would affect the current results. Additionally, approximately 30% of patients who agreed to participate did not complete the questionnaire for unclear reasons. Although this percentage seems high, it is within the range for achieving our objectives in future studies, considering the sensitive nature of some questions. The decision algorithm must be tested in a larger population to assess its good performance.

Our study had several strengths. We employed a systematic Delphi expert consensus methodology which relied on a previously validated analogic questionnaire updated to the current testing guidelines and adapted into a user-friendly digital platform. With a limited number of participants, we were able to diagnose one undiagnosed HIV patient and refer nine participants to the PrEP program. If the online tool proves successful in real-life settings, it could be integrated into the electronic health record systems used by primary care providers, ensuring the long-term sustainability of this strategy beyond the study phase.

## 5. Conclusions

An online, self-administered questionnaire that assesses HIV and HCV exposure risk and indicator conditions successfully identifies patients already infected with HIV and HCV, as well as those in a PrEP program. Responders mostly find the use of such a questionnaire easy to use, and implementation at a grander scale can be tested to try to overcome patient and physician barriers to timely testing.

## Figures and Tables

**Table 1 biomedicines-12-01846-t001:** Participants’ demographics and questionnaire output, by group.

Participants	Attending Primary Care (*n* = 41)	With a Recently Acquired HIV Infection (*n* = 10)	With HIV Infection Acquired > 6 Months before (*n* = 20)	On a PrEP Program (*n* = 21)	With an Active HCV Infection (*n* = 10)
Total participants, *n* (%)	41 (40.2)	10 (9.8)	20 (19.6)	21 (20.6)	10 (9.8)
Sex at birth, male, *n* (%)	21 (51.2)	10 (100)	20 (100)	21 (100)	8 (80)
Age, mean (SD)	38.4 (13.5)	32.2 (7.1)	40.5 (10.9)	36 (7.2)	57.4 (7.7)
Country of origin, *n* (%)					
Spain	31 (75.6)	1 (10)	11 (55)	13 (61.9)	9 (90)
Latin America	8 (19.5)	9 (0)	6 (30)	7 (33.3)	1 (10)
Other	2 (4.9)	9 (0)	3 (15)	1 (4.8)	0 (0)
Education level college or greater, *n* (%)	38 (92.7)	9 (90)	19 (95)	21 (100)	5 (50)
Working situation, currently active, *n* (%)	36 (87.8)	9 (90)	16 (80)	18 (85.7)	2 (20)

Abbreviations: SD, standard deviation; HIV, human immunodeficiency virus; HCV, hepatitis C virus; PrEP, HIV pre-exposure prophylaxis.

**Table 2 biomedicines-12-01846-t002:** Questionnaire output.

Participants	Attending Primary Care (*n* = 41)	With a Recently Acquired HIV Infection (*n* = 10)	With HIV Infection Acquired > 6 Months before (*n* = 20)	On a PrEP Program (*n* = 21)	With an Active HCV Infection (*n* = 10)
HIV/HCV testing indication, *n* (%)	26 (63.4)	10 (100)	19 (95)	21 (100)	10 (100)
PrEP derivation indicated, *n* (%)					
No	32 (78.1)	0 (0)	3 (15)	0 (0)	8 (80)
Individual assessment required	3 (7.3)	6 (60)	6 (30)	2 (9.5)	1 (10)
Direct derivation	6 (14.6)	4 (40)	11 (55)	19 (90.5)	1 (10)

Abbreviations: HIV, human immunodeficiency virus; HCV, hepatitis C virus; PrEP, HIV pre-exposure prophylaxis.

**Table 3 biomedicines-12-01846-t003:** Patients’ evaluation of the questionnaire.

Participants	Attending Primary Care (*n* = 41)	With a Recently Acquired HIV Infection (*n* = 10)	With HIV Infection Acquired > 6 Months before (*n* = 20)	On a PrEP Program (*n* = 21)	With an Active HCV Infection (*n* = 10)
Time to finish the questionnaire in minutes, median (IQR)	5.7 (4.1–8.7)	5.9 (4.7–9.6)	6.5 (5.2–8.9)	7.0 (4.7–10.6)	8.9 (6.9–11.9)
Were the questions easy to answer? Yes, *n* (%)	40 (97.6)	9 (90)	20 (100)	21 (100)	10 (100)
Did you need help to finish it? Yes, *n* (%)	4 (9.8)	1 (10)	1 (5)	0 (0)	2 (20)
Do you think the tool is easy to use? Yes, *n* (%)	40 (97.6)	10 (100)	20 (100)	21 (100)	9 (90)
Would you recommend the tool to other users? Yes, *n* (%)	40 (97.6)	10 (100)	20 (100)	20 (95.2)	10 (100)
How did you complete the questionnaire? *n* (%)					
Personal mobile device	37 (90.2)	10 (100)	19 (95)	21 (100)	9 (90)
Personal computer	3 (7.3)	0 (0)	0 (0)	0 (0)	0 (0)
A member of the healthcare team helped me	1 (2.4)	0 (0)	0 (0)	0 (0)	1 (19)
Other	0 (0)	0 (0)	1 (5)	0 (0)	0 (0)

Abbreviations: IQR, interquartile range; HIV, human immunodeficiency virus; HCV, hepatitis C virus; PrEP, HIV pre-exposure prophylaxis.

## Data Availability

All data presented in this study can be obtained through reasonable requests from the corresponding authors.

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
