# Peer review of "Design and Proof of Concept of a Web-Based Questionnaire to Identify Patients at Risk for HIV and HCV Infection"

_biomedicines, 2024, doi:10.3390/biomedicines12081846_

Round 1

Reviewer 1 Report (Previous Reviewer 2)

Comments and Suggestions for Authors

The main purpose of the presented questionnaire is to identify patients at high risk of HCV or HIV infection. However, the study is based on a sample of patients already diagnosed. The use of the questionnaire in such a biased sample has nothing to do with the identification of infected individuals. For the purpose of identifying infected people, it is necessary to apply the questionnaire proposed by the authors to a mixed population of people, including both infected and healthy people. This has not been done. Consequently, the aim of the work has not been achieved.

In the introduction of the article, the authors also formulated the goal of the work as improving the accuracy of diagnosis of infected patients. This implies a comparative study, which was not performed.

Despite the fact that the authors answered that they performed a thorough statistical analysis - they used for this purpose data from other work, obtained with other questionnaires. The validity of such statistical analysis is highly questionable.

Only a secondary objective of the questionnaire has been accomplished - improving its usability, adapting it to a web version to increase the number of people able to complete it to the end. But this lacks any scientific and has only a small applied value. The work is still just a preliminary study, with serious design flaws that do not allow an objective assessment of the advantages and disadvantages of the created questionnaire even for a small sample size.

Author Response

Reviewer 2 Report (Previous Reviewer 3)

Comments and Suggestions for Authors

Authors have addressed the problems and revised the manuscript.

Author Response

We would like to thank the reviewer for the time spent and the previous comments to improve the quality of our manuscript.

Reviewer 3 Report (Previous Reviewer 4)

Comments and Suggestions for Authors

Authors have added sensitivity and specificity values.

and amended the Title.

Article is acceptable for publication.

Author Response

We would like to thank the reviewer for the time spent, the positive evaluation of this work, and the recommendation for publication of our manuscript.

Round 2

Reviewer 1 Report (Previous Reviewer 2)

Comments and Suggestions for Authors

The authors have correctly stated the purpose and status of their work as described in the current version of the manuscript, which removes most questions.

There is only one minor comment. Please add statistical analysis of the data to the Supplementary.

Author Response

We would like to thank the reviewer for his time in reviewing our manuscript. 

We have added the statistical analysis to the Supplementary Material as Annex 3, as requested.

This manuscript is a resubmission of an earlier submission. The following is a list of the peer review reports and author responses from that submission.

Round 1

Reviewer 1 Report

Comments and Suggestions for Authors

General overview

- The paper is about a tool but lacks the presentation of its algorithm.

- You must be a speaker of a specific language to use the tool but this is not presented anywhere.

- The applied methods are not presented in sufficient detail to allow the reproduction of the study.

- Sensitive data are collected (e.g., name of participant) but this is not highlighted anywhere.

- The pool of respondents is imbalanced and apparently do not reflect the target population.

General comments

- Abbreviations are define first time when are used in the abstract and in the body of the manuscript.

- Do not end the titles and subtitles with the full stop.

- Do not use abbreviations in the title of the manuscript.

- Do not start a new paragraph with "However".

- Do not start a sentence with "This"/"These", and abbreviation, a number.

Abstract

- Delete "Background:", "Methods:", "Results:", "Conclusions: "

- State the aim of the study.

- It is unclear which is the link between HIV and HCV, it is about HCV on HIV positive patients?

- It is unclear why a Delphi method was used on a validated questionnaire.

- When the study was conducted?

Introduction

- "remains a significant public health challenge" where?

- "Some recent studies" is not supported by only one reference.

- "a web-based tool" for what? screening? diagnosis?

Methods

- If I correctly understood, you created a decision support system, not a questionnaire.

- "we recruited 19 skilled physicians" how? from where? based on which criteria? etc. You must fully describe the process.

- Why do you have an unbalance between gastroenterologists and specialists in HIV and STIs?

- " They provided anonymous opinions " online? on paper?

- Define "ewer experts".

- Present how the algorithm was created.

- "provided to one primary care provider at each primary care centre" It is confusing; please rephrase for clarity.

- The number of respondents is a result not a method. The inclusion and exclusion criteria are unclear. It is also unclear how the participants were selected.

- Why 100+?

Results

- "78.4% were men" this result does not reflect the general population. The same comment for " 92% had a high educational level".

- The sample size is small, please provide the report (e.g., 1/10).

Discussion

- Do not repeat in this section the information already presented in the manuscript.

- Begin the discussion by briefly summarizing your main findings.

- Explore possible mechanisms or explanations for your findings.

- Emphasize the new and important aspects of your study and put your findings in the context of the totality of the relevant evidence.

- State the limitations of your study, and explore the implications of your findings for future research and for clinical practice or policy.

- Discuss the influence of the respondent's characteristics on your findings.

- Discuss the generalizability of your finding.

- Discuss the practical utility of your findings.

- Do not repeat in detail data or other information given in other parts of the manuscript, such as in the Introduction or the Results section.

Conclusion

- List in this section only the conclusions supported by your findings.

Author Response

General overview

1- The paper is about a tool but lacks the presentation of its algorithm. The applied methods are not presented in sufficient detail to allow the reproduction of the study.

   Authors: We thank Reviewer #1 for the comments to improve our manuscript. To address the Reviewer’s concern regarding the algorithm and the reproductivity of the study, we have expanded the Methods section to further explain the decision algorithm. The algorithm as it is extracted from REDCap is presented in the Supplementary Material, Appendix 2. The complete questionnaire with the initial and final questions is available in Supplementary Material, pages 2-3 and 9-13. The responses of each Delphi participant are available in the Supplementary Material, pages 2-3. The design and aim of the pilot are explained in the Methods section and can now be replicated.

2- You must be a speaker of a specific language to use the tool but this is not presented anywhere.

   Authors: We agree that the tool is in Spanish; therefore, you have to be a Spanish speaker, so we have reflected this in the manuscript (line 88).

3- Sensitive data are collected (e.g., name of participant) but this is not highlighted anywhere.

   Authors: Regarding sensitive data, like the participant's name, these data are stored in the RedCAP tool, hosted at the servers of Ramón y Cajal Hospital, inside the secure network of Madrid’s health system. Every participant who is included signed an electronic informed consent to participate, which is also hosted in the same secure servers. The Ethics Committee of our Hospital and Madrid’s Primary care network approved the study. Researchers only extracted anonymized data from the servers . We have reflected this in the methods section of the manuscript (lines 167,168 and 174,175)

4- The pool of respondents is imbalanced and apparently do not reflect the target population.

   Authors: The population of the pilot study does not intend to reflect the general population; it rather mimics the current shape of the HIV and HCV pandemic in Spain, where most people are male, MSM and with high educational levels. People with recent HIV infections or on a PrEP program are mainly male. Among the people who were not positive controls and were recruited at primary care centres, 49.8% were women. Besides, in Spain, HIV and HCV occur predominantly among males (88.6%), MSM (72.6%), highly educated (60%), as can be found in most of the cohort studies published in our country (García-Ruiz de Morales et al. Lancet HIV 2024, https://doi.org/10.1016/S2352-3018(24)00118-8)

General comments

5- Abbreviations are define first time when are used in the abstract and in the body of the manuscript.

6- Do not end the titles and subtitles with the full stop.

7- Do not use abbreviations in the title of the manuscript.

8- Do not start a new paragraph with "However". Do not start a sentence with "This"/"These", and abbreviation, a number.

Authors: We have made minor corrections, to address the Reviewer’s recommendations.

Abstract

9- Delete "Background:", "Methods:", "Results:", "Conclusions: "

Authors: We have deleted it as suggested.

10- State the aim of the study.

Authors: We have stated the aim of the study in the abstract (lines 37-38).

11- It is unclear which is the link between HIV and HCV, it is about HCV on HIV positive patients?

Authors: Due to similarity in risk factors and transmission, and the high prevalence of HCV/HIV coinfection, an overlap in screening services has been proposed. We have further explained it in the Methods section. The ECDC guidance on integrated testing of hepatitis B (HBV), hepatitis C (HCV) and HIV supports countries in the global effort to combat viral hepatitis and eliminate HIV as public health threats by 2030 (https://www.ecdc.europa.eu/en/publications-data/public-health-guidance-hiv-hepatitis-b-and-c-testing-eueea).

12- It is unclear why a Delphi method was used on a validated questionnaire.

Authors: The Delphi method was used to ensure that adapting the previously validated questionnaire (validated in 2016) to the most recent HIV and HCV testing and HIV pre-exposure prophylaxis (PrEP) guidelines did not modify its validity.

13- When was the study conducted?

Authors: We have explained when the study was conducted (line 42).

14- Introduction

 "remains a significant public health challenge" where?

 "Some recent studies" is not supported by only one reference.

 "a web-based tool" for what? screening? diagnosis?

Authors: We have made minor corrections to the Introduction to clarify the Reviewer’s concerns (lines 58, 65, 87).  

Methods

15- If I correctly understood, you created a decision support system, not a questionnaire.

Authors: We agree with the Reviewer that we have created a decision support system based on a questionnaire. We created a new questionnaire from the previously validated one to also include PrEP guidelines and the updated testing guidelines. The objective of the study was to adapt the previously validated one to be able to use it in an electronic tool. The objective of this phase of the study is not to validate the decision support system, but to assess the feasibility of implementing such electronic questionnaire at a primary care setting. The calibration of the predictive model will be done at the upcoming phase, where we aim to administer more than 10,000 questionnaires to general population who attend a primary care centre. The current algorithm is based on the previously validated questionnaire and has been modified only to incorporate the current indications for PrEP derivation. In order to address the Reviewer’s concern, we have further expanded the limitations section (lines 289-290).

16- "we recruited 19 skilled physicians" how? from where? based on which criteria? etc. You must fully describe the process.

            Authors: We selected the experts based on predefined selection criteria that are specified in Supplementary Material page 2 (lines 25-44).

17- Why do you have an unbalance between gastroenterologists and specialists in HIV and STIs?

            Authors: Most of the questions are directed at identifying people at risk of HIV, and most of the people who treat HIV also treat HIV-HCV coinfections. We decided to include some gastroenterologists as well, just in case they could provide some better ideas on questions to assess HCV mono-infection better.

18- " They provided anonymous opinions " online? on paper?

     Authors: The opinions were provided online, and we have reflected so in the manuscript (line 114).

19- Define "ewer experts".

      Authors: We understand that the Reviewer had an incomplete manuscript version. We think the Reviewer refers to line 121, which explains the responses of the Delphi participants: “if five or fewer experts suggested an alternative to ‘Essential’, the response was to indicate agreement to maintain the question”.

20- Present how the algorithm was created.

      Authors: As stated in response #17, we created a new questionnaire from the previously validated and published questionnaire to also include the PrEP guidelines and updated guidelines on HIV and HCV screening in our setting. As mentioned in response #1, the algorithm as extracted from REDCap is presented in the Supplementary Material, Appendix 2. The complete questionnaire with initial and final questions is available in Supplementary Material, pages 2-3 and 9-13. The responses of each Delphi participant are available in the Supplementary Material, pages 2-3. The design and purpose of the pilot test are explained in the Methods section and can now be reproduced. We have expanded the Methods section to explain the decision algorithm in more detail.

21- "provided to one primary care provider at each primary care centre" It is confusing; please rephrase for clarity.

      Authors: We have rephrased for clarity (line 169).

22- The number of respondents is a result not a method. The inclusion and exclusion criteria are unclear. It is also unclear how the participants were selected. Why 100+?

            Authors: We have modified the wording as suggested (lines 167, 168). In the methods section we only explain that we wanted to include at least 100 participants. The actual results are in the Results section. We have clarified the inclusion and exclusion criteria in the Methods section (lines 171-172).

Results

23- "78.4% were men" this result does not reflect the general population. The same comment for " 92% had a high educational level".

            Authors: We agree with the Reviewer on this comment. Nevertheless, the population of the pilot study does not intend to reflect the general population, it rather mimics the current shape of the HIV and HCV pandemic in Spain, where most people are male, MSM and with high educational level. People with recent HIV infections on a PrEP program are mainly male, but among the people who were not positive controls and were recruited at primary care centers, 49.8% were women.

24- The sample size is small, please provide the report (e.g., 1/10).

            Authors:  We have provided the report throughout the Results section.

25- Discussion

- Do not repeat in this section the information already presented in the manuscript. Begin the discussion by briey summarizing your main ndings.

- Explore possible mechanisms or explanations for your ndings.

- Emphasize the new and important aspects of your study and put your findings in the context of the totality of the relevant evidence.

- State the limitations of your study, and explore the implications of your ndings for future research and for clinical practice or policy.

- Discuss the influence of the respondent's characteristics on your findings.

- Discuss the generalizability of your finding.

- Discuss the practical utility of your findings.

- Do not repeat in detail data or other information given in other parts of the manuscript, such as in the Introduction or the Results section.

Conclusion

- List in this section only the conclusions supported by your findings.

            Authors: We have revised the Discussion and Conclusions sections to comply with the guidelines the Reviewer has recommended.

Reviewer 2 Report

Comments and Suggestions for Authors

The idea of the study is very important and interesting and can help humanity in the fight against HIV and HCV pandemics. The authors have developed a user-friendly questionnaire that can identify a person's risk group for HIV and HCV and refer them for timely testing. This may help to solve the problem of late diagnosis of these diseases. Despite the high significance of the idea and the attempt to proof it, this paper has serious flaws that make the manuscript unacceptable for publication in its current state.

Major comments

This paper lacks novelty compared to previous papers published by these authors (DOI: 10.1371/journal.pone.0220375, DOI: 10.1111/jvh.13413). Questionnaire simplification is a two-edged sword: on the one hand, it makes it more convenient and faster to test people, but on the other hand, it creates the risk of increasing false negatives. Therefore, a comparative study of original and short questionnaires is necessary.

This work does not have a control group, so it lacks the most important characteristic of the questionnaire - its specificity. Moreover, as in any other clinical test, analysis of the AUC curve is necessary to evaluate the key characteristics of the test. When screening groups of patients, we naturally expect a large number of positive identifications, but this is "searching by flashlight".

I am concerned that the sample is predominantly male, and in 3 of the 5 groups they make up 100% of the study population. Women should be added as they are also at high risk of HIV and HCV infection, for example by being partners of bisexual men.

In addition, the sample is biased in terms of the part of society being studied. Most patients have higher education and are employed. The problem of HIV, hepatitis C and other similar infections is probably more acute among non-working people without higher education (who tend to be less health-conscious and aware of the risks). Therefore, for this program to truly address the HIV and hepatitis C pandemic, more people without higher education and without jobs need to be included.

In order to obtain statistically significant results, the sample size needs to be significantly increased. The previous works of the authors can be considered as pilot studies. The next logical step is for the authors to verify their questionnaire on a large sample size.

Since the questionnaire contains socially undesirable questions, this will inevitably lead to subjects lying in their answers. A system of additional questions assessing the level of lying in response to these questions should be considered. It is necessary to think about a system of questions formulated in such a way as not to provoke negative reactions from test takers and to ensure a true answer. Therefore, in developing a really good questionnaire, data from the psychological analysis of risk groups should be involved.

Two participants found the test difficult, and a fairly large percentage needed help in completing it. It is necessary to find out the reason for the difficulty and correct these questions (e.g., add explanations). Making the questions more accessible is inevitable so that the questionnaire can be tested with less-educated risk groups.

The development of an effective questionnaire also requires careful validation of its structure. In this respect, it is comparable to validating the structure of psychological questionnaires. No validation of the structure of the questionnaire has been performed by the authors.

Author Response

The idea of the study is very important and interesting and can help humanity in the fight against HIV and HCV pandemics. The authors have developed a user-friendly questionnaire that can identify a person's risk group for HIV and HCV and refer them for timely testing. This may help to solve the problem of late diagnosis of these diseases. Despite the high significance of the idea and the attempt to proof it, this paper has serious flaws that make the manuscript unacceptable for publication in its current state.

Major comments

1- This paper lacks novelty compared to previous papers published by these authors (DOI: 10.1371/journal.pone.0220375, DOI: 10.1111/jvh.13413).

Authors: We would like to thank Reviewer #2 for his thorough evaluation of our manuscript and her/his concern regarding the HIV pandemic. We would like to explain that our two previous papers on this matter, cited by the Reviewer, differ significantly from the intended approach of the current study. First, the questionnaires in the other two studies were analogical, and required of the presence of a healthcare professional to complete them. Secondly, there were specific nurses employed within the study who performed the rapid diagnostic tests. These two facts make previous studies not generalisable and applicable to a broader number of participants. In this new study, we started from the basis and the decision algorithm of our previously validated questionnaire and performed a Delphi questionnaire to include new questions to reflect the most updated guidelines on HIV, HCV testing and PrEP derivation. The current online self-administered questionnaire aims to allow people who attend primary care to be able to auto-screen themselves with the questionnaire, with no need to contact a healthcare professional. We understand this could help address some of the main barriers for HIV and HCV testing.

2- Questionnaire simplification is a two-edged sword: on the one hand, it makes it more convenient and faster to test people, but on the other hand, it creates the risk of increasing false negatives. Therefore, a comparative study of original and short questionnaires is necessary.

Authors: The aim of the current study is not to simplify the questionnaire, but rather to be able to adapt it into an online tool. In fact, the initial questionnaire had 21 questions and the current one, as it can be consulted in the Supplementary material, has more than 35 (Supplementary Material pages 9-13). The objective of this phase of the study is not to validate the questionnaire, but to assess the feasibility of implementing such electronic questionnaire at a primary care setting. The calibration of the model will be done at the upcoming phase, where we aim to administer more than 10,000 questionnaires to general population who attend a primary care centre.

3- This work does not have a control group, so it lacks the most important characteristic of the questionnaire - its specificity. Moreover, as in any other clinical test, analysis of the AUC curve is necessary to evaluate the key characteristics of the test. When screening groups of patients, we naturally expect a large number of positive identifications, but this is "searching by flashlight".

Authors: We agree with the Reviewer on this point. This pilot study does not have a control group, and therefore, validity and specificity cannot be ascertained. As we state throughout the study, the objective of this phase of the study is only to assess feasibility at the primary care level (with the primary care population included) and to ensure that it adequately identifies people who are already infected with HIV, HCV, or PrEP. Finally, we wanted to know what the participants thought regarding the feasibility and ease of the questionnaire. Therefore, we used a questionnaire previously validated for HIV and HCV screening, and its sensitivity and negative predictive value had already been calculated (100% for both). The relevant outcome of the questionnaire is the need for testing or need for PrEP derivation. Given these outcomes, as the Reviewer states, there are a high number of positive questionnaires, which will ensure that the population that needs to be screened will be screened if they complete the questionnaire. In the positive controls, who were the people who have already been diagnosed with HIV or who are already in PrEP, the questionnaire performs excellent and would have identified for testing all people who were recently infected of HIV, all patients on PrEP and all patients with an active HCV infection. The questionnaire would have only missed for testing one HIV patient who had been diagnosed several years ago, because he didn’t report unprotected sex, more than one partner, drug use, STI or other HIV indicator conditions.

As stated in the previous response, the analysis of the predictive model (sensitivity, specificity, AUC, predictive values…) will be done at the upcoming phase, where we aim to administer more than 10,000 questionnaires to general population who attend a primary care centre. We have, nevertheless, stated as a limitation of our study that Sensitivity, Specificity and predictive values cannot be calculated (lines 283-285).

4- I am concerned that the sample is predominantly male, and in 3 of the 5 groups they make up 100% of the study population. Women should be added as they are also at high risk of HIV and HCV infection, for example by being partners of bisexual men.In addition, the sample is biased in terms of the part of society being studied. Most patients have higher education and are employed. The problem of HIV, hepatitis C and other similar infections is probably more acute among non-working people without higher education (who tend to be less health-conscious and aware of the risks). Therefore, for this program to truly address the HIV and hepatitis C pandemic, more people without higher education and without jobs need to be included.

            Authors: We would like to thank the reviewer for his comment. The population of the pilot study does not intend to reflect the general population; it rather mimics the current shape of the HIV and HCV pandemic in Spain, where most people are male, MSM and with high educational levels. People with recent HIV infections or on a PrEP program are mainly male. Among the people who were not positive controls and were recruited at primary care centres, 49.8% were women. Besides, in Spain, HIV and HCV occur predominantly among males (88.6%), MSM (72.6%), highly educated (60%), as can be found in most of the cohort studies published in our country (García-Ruiz de Morales et al. Lancet HIV 2024, https://doi.org/10.1016/S2352-3018(24)00118-8). We agree that the use of an online tool requires a certain level of education, and the tool probably will under-screen people with lower levels of education. We have expanded the limitations of our work to reflect this issue (lines 284-286).

5- In order to obtain statistically significant results, the sample size needs to be significantly increased. The previous works of the authors can be considered as pilot studies. The next logical step is for the authors to verify their questionnaire on a large sample size.

            Authors: We agree with the Reviewer. We did not calculate any p-values for this study because the aim was only to test its feasibility and accuracy in diagnosing HIV or HCV in previously positive patients. We are starting the following phase of the study, where the questionnaire will be fully available for more than 2 years at different primary care centers and we will aim to see if the availability of the questionnaire improves the rates of HIV and HCV testing and diagnosis.

6- Since the questionnaire contains socially undesirable questions, this will inevitably lead to subjects lying in their answers. A system of additional questions assessing the level of lying in response to these questions should be considered. It is necessary to think about a system of questions formulated in such a way as not to provoke negative reactions from test takers and to ensure a true answer. Therefore, in developing a really good questionnaire, data from the psychological analysis of risk groups should be involved.

            Authors: We agree with the Reviewer in this issue, which is one of the main gaps for HIV and HCV testing. Nevertheless, the implementation of the questionnaire tries to address exactly that: the fact that many individuals don’t feel comfortable on talking about their sexual practices with their physicians and might lie to them. This can always also be the case in a questionnaire, but we understand that it will improve compared to speaking to the physician. Nevertheless, the aim of the questionnaire is to see if its implementation would improve HIV and HCV testing compared to not using it. If it doesn’t, this could be one of the explanations. We have further extended the limitations to include this fact (line 281). The answers were confidential for the primary care physicians who are responsible of the patient and only the alert is sent to them for HIV/HCV testing of PrEP derivation.

7- Two participants found the test difficult, and a fairly large percentage needed help in completing it. It is necessary to find out the reason for the difficulty and correct these questions (e.g., add explanations). Making the questions more accessible is inevitable so that the questionnaire can be tested with less-educated risk groups.

Authors: Due to the design of the questionnaire, we are not able to identify what made the questionnaire difficult to complete to these two participants. Nevertheless, they finished the questionnaire with some external help. We completely agree that accessibility to the less-educated risk groups will be a problem of the questionnaire, despite the fact that most people, educated or not, are nowadays comfortable with the use of a online tools. In these vulnerable populations, the intervention of a primary care provider will probably still be necessary to successfully screen for HIV and HCV. Despite this fact, only 2 out of 102 patients found the questionnaire difficult to finish, and only 6 out of 102 needed some help to finish it. We consider these numbers reasonable: we aim to  improve the usual HIV and HCV screening, by simplifying primary care physician’s identification of testing conditions. With this data, more than 94% of the participants would have been “screened” with the questionnaire without the need of external help. If a participant requires help to finish the questionnaire, we think it will be less time consuming for the health provider to help them rather than to ask all the questions and individually assess for HIV and HCV testing indications. The questionnaire never intends to replace the physician’s need for screening, but to help them and reduce the time to be able to perform the risk and indicator condition assessment.

8- The development of an effective questionnaire also requires careful validation of its structure. In this respect, it is comparable to validating the structure of psychological questionnaires. No validation of the structure of the questionnaire has been performed by the authors.

            Authors: The aim of this phase of the study was only to assess its feasibility, possibilities of implementation at primary care level and the correct identification of infected patients and patients to be referred to PreP. At this point, the structure has been careful designed to assure that inquires all relevant issues that guidelines (HIV-HCV Testing and PreP) contemplate for the purposes of the questionnaire. In the following phase of the study, where we aim to include more than 10,000 participants, we will be able to validate many aspects that are not possible right now, as the implementation part of the study does not have enough power for this validation.

Reviewer 3 Report

Comments and Suggestions for Authors

In this study, authors designed a web-based questionnaire to identify patients at risk for HIV and HCV infection at the primary care level. A total of 142 participants provided informed consent, and 102 completed the questionnaire. The topic is interesting, but some issues should be considered.

1.         Full names should be provided when the abbreviations appeared for the first time. For example, HIV and HCV in the Abstract.

2.         I suggested that authors can provide more information about the late diagnosis of HIV and HCV information in the Introduction.

3.         The main issue is the small number of users enrolled in the pilot study. Do authors suppose this would limit the significance of this study?

4.         Proofreading is suggested.

Comments on the Quality of English Language

Minor editing of English language required

Author Response

In this study, authors designed a web-based questionnaire to identify patients at risk for HIV and HCV infection at the primary care level. A total of 142 participants provided informed consent, and 102 completed the questionnaire. The topic is interesting, but some issues should be considered.

            Authors: We would like to thank Reviewer #3 for the positive evaluation of our work. 

1- Full names should be provided when the abbreviations appeared for the first time. For example, HIV and HCV in the Abstract.

Authors: We have revised the abbreviations throughout the manuscript.

2- I suggested that authors can provide more information about the late diagnosis of HIV and HCV information in the Introduction.

Authors: We have expanded the Introduction to include the magnitude of the problem (lines 65-67).

3- The main issue is the small number of users enrolled in the pilot study. Do authors suppose this would limit the significance of this study?

Authors: We agree that there are only a limited number of participants included. Nevertheless, the aim of this phase of the study was only to adapt the analogical tool to a more updated online one that the participant can auto-administer, and to see if its implementation at greater scale makes sense and is feasible. The study tries to set the bases for the next phase where we are testing the questionnaire in 8 different primary care centers for 2 years, to see if the availability of the questionnaire increments HIV and HCV testing compared to the current standard of diagnosis with risk and indicator conditions identification by primary care physicians.

4- Proofreading is suggested.

Comments on the Quality of English Language

Minor editing of English language required

Authors: We have further revised the article to avoid any misspellings or errors.

Reviewer 4 Report

Comments and Suggestions for Authors

I have reviewed the paper by Garcia-Ruiz

Although the surveyed cohort is relatively small for such an important aim, it is sufficient enough to test their hypothesis and provide proof of concept. Authors appropriately call it pilot implementation.

The questionnaire is not entirely original, as it is derived from previous existing ones. I am unsure of 19 physicians input is a good idea, and it seemed rather excessive for coauthorship.

If that amount of questions is too much or feasible to be utilized in the real patient world will be known once this is implemented in their hospital.

75.6% of patients were Spanish. Authors need to specify that the rest (almost 25%) were fluent Spanish speaking.

Analysis presented is so far descriptive. Sensitivity and specificity values, a concordance analysis is necessary.

Author Response

I have reviewed the paper by Garcia-Ruiz

1- Although the surveyed cohort is relatively small for such an important aim, it is sufficient enough to test their hypothesis and provide proof of concept. Authors appropriately call it pilot implementation.

            Authors: We would like to thank Reviewer #4 for the positive assessment of our work. 

2- The questionnaire is not entirely original, as it i derived from previous existing ones.

Authors: As stated by the Reviewer, the questionnaire was not entirely original, as we based the new online tool on our previously validated analog questionnaire.

3- I am unsure of 19 physicians input is a good idea, and it seemed rather excessive for coauthorship.

Authors: Not everybody who participated in the DELPHI is included as co-authors, as they are listed in the ATENEA-development group. The authors listed are the ones that designed the initial questionnaire and the study and those who recruited patients for the pilot study. We are open, nevertheless, to reduce the number of authors if needed.

4- 75.6% of patients were Spanish. Authors need to specify that the rest (almost 25%) were fluent Spanish speaking.

Authors: We have added a small clarification that all participants indeed were fluent spanish speakers, as most of them were originally from Latin America.

5- If that amount of questions is too much or feasible to be utilized in the real patient world will be known once this is implemented in their hospital. Analysis presented is so far descriptive. Sensitivity and specificity values, a concordance analysis is necessary.

            Authors: We agree that the data right now is descriptive, as the aim of the current phase of the work is just to test feasibility and possibility of implementation at a greater scale. We are currently starting the main work, where our aim will be to include more than 10,000 participants to be able to assess if the availability of the questionnaire improves HIV and HCV testing, and to validate the sensitivity and specificity of the questionnaire. Nevertheless, specificity, sensitivity and predictive values were validated for the previous questionnaire, and most of the algorithm was like the one in that previous questionnaire.

Round 2

Reviewer 1 Report

Comments and Suggestions for Authors

Although most of my comments and suggestions were appropriately addressed, some major issues remain.

- The title misleads the readers that the study is conducted at the primary care level, but the participants in the study do not reflect the primary care population. "the primary care level" should not be in the title.

- The paragraph that starts in line 76 should be linked to the previous paragraph.

- I do not see the role of "primary care" in the aim of the study.

- Define " CDC", and "ECDC" abbreviations.

- Lines 126-140 are Results not methods.

- It is still unclear how the "primary care provider " was selected.

- "40 patients from the target primary care population (five from each of the eight assigned primary care centres at Ramón y Cajal Hospital), 20 HIV patients diagnosed between six months and ten years ago (positive controls for HIV testing), ten patients newly diagnosed with HIV (positive controls for HIV testing and PrEP), and ten patients in the PrEP program (positive controls for HIV testing and PrEP". It is still unclear how did you reach these numbers.

- Define "their criteria".

- I did not capture the evaluation of accuracy.

- I expected that all answers to my previous comments and suggestions to be included in the manuscript.

Reviewer 2 Report

Comments and Suggestions for Authors

The authors have answered the questions but unfortunately have not addressed the serious flaws of the study.

 In its present state, the manuscript describes a preliminary study that lacks critical steps in its design to be logically complete and have scientific and clinical significance.

The structure of the questionnaire is not validated and adapted for less educated people.

 The results of the study are trivial and are not supported by accurate and rigorous statistical analysis.

Reviewer 4 Report

Comments and Suggestions for Authors

Since my initial comments were not satisfactorily address, I am going to be very direct this time. Authors state in their reply thet "The authors listed are the ones that designed the initial questionnaire and the study and those who recruited patients for the pilot study." 

So, again, I do not think an input of 19 physicians is a good for a questionnaire design. (Too many cooks in the kitchen). They need to trim down the excessively long list of co-authors, and only keep those who actively participated in the study and elaboration of the manuscript. 

Authors need to perform and present sensitivity and specificity values, and a concordance analysis despite their small cohort. The paper is a "proof of concept" so they need to prove the validity.